behaviour/biomechanics/health and disease and epidemiology

insect flight, parasite, skill learning,
*Apis mellifera*, manoeuvrability, control

**Author for correspondence:**
Florian T. Muijres
e-mail: florian.muijres@wur.nl

# *Varroa destructor* infestation impairs the improvement of landing performance in foraging honeybees

Florian T. Muijres[1], Coby van Dooremalen[2],
Martin Lankheet[1], Heleen Lugt[1], Lana J. de Vries[1,2,3,4]
and Frank Van Langevelde[3]

[1]Experimental Zoology Group, [2]Bees@WUR, [3]Wildlife Ecology and Conservation Group, and [4]Behavioural Ecology Group, Wageningen University & Research, Wageningen, The Netherlands

FTM, 0000-0002-5668-0653; CvD, 0000-0002-1525-0171;
ML, 0000-0002-8261-2187; FVL, 0000-0001-8870-0797

The parasitic mite *Varroa destructor* is an important contributor to the high losses of western honeybees. Forager bees from *Varroa*-infested colonies show reduced homing and flight capacity; it is not known whether flight manoeuvrability and related learning capability are also affected. Here, we test how honeybees from *Varroa*-infested and control colonies fly in an environment that is unfamiliar at the beginning of each experimental day. Using stereoscopic high-speed videography, we analysed 555 landing manoeuvres recorded during 12 days of approximately 5 h in length. From this, we quantified landing success as percentage of successful landings, and assessed how this changed over time. We found that the forager workforce of *Varroa*-infested colonies did not improve their landing success over time, while for control bees landing success improved with approximately 10% each hour. Analysis of the landing trajectories showed that control bees improved landing success by increasing the ratio between in-flight aerodynamic braking and braking at impact on the landing platform; bees from *Varroa*-infested colonies did not increase this ratio over time. The *Varroa*-induced detriment to this landing skill-learning capability might limit forager bees from *Varroa*-infested colonies to adapt to new or challenging conditions; this might consequently contribute to *Varroa*-induced mortality of honeybee colonies.

# 1. Introduction

The western honeybee (*Apis mellifera*) is an important pollinator, both in nature and in agriculture. Apiculture and crop pollination are seriously affected by widespread losses of honeybee colonies [1,2]. Causes for these colony losses are diverse and controversial, but exposure to the parasitic mite *Varroa destructor* has been suggested as one of the major contributing factors [3]. The mites parasitize both adult bees and the brood and also function as a vector for several diseases, such as the deformed wing virus [4]. Drainage of reserves and viral diseases can put the whole colony under stress, which can eventually lead to colony collapse.

*Varroa destructor* infestation of a honeybee colony impairs performance of foragers. It results in a more prolonged absence from the colony during foraging trips and reduced homing success [5], most likely due to reduced non-associative learning [6]. Workers begin foraging at younger ages when they have been infested by *V. destructor* as pupae, hence affecting temporal task schedules [7,8]. In addition, infested foragers carry smaller pollen loads back to the colonies resulting in a higher recruitment of pollen foragers, but not enough to prevent a smaller brood nest (and hence a smaller workforce) at the end of the season [9]. Moreover, with the same fuel loads, bees from *Varroa*-infested colonies fly shorter distances than those from control colonies [10]. In addition, the deformed wing virus can cause severe wing deformation that leads to flight inability. Active foragers without visual signs of the disease but clinically relevant deformed wing virus infections [11] show reduced flight duration and distance [12,13], similar to the effect of *V. destructor* [10].

Optimal foraging performance requires good manoeuvrability and precise landing control. While visiting flowers, forager bees need to navigate through complex environments [14], and perform numerous sequential take-off and landing manoeuvres on flowers and the hive entrance [15]. A single foraging trip may involve more than 1000 landings [16]. Moreover, the targets for landing may change from day to day, when new foraging resources are exploited. Bees, therefore, need to be able to adapt their manoeuvrability and landing control. If *Varroa* infestation would impair flight control or the capability to learn manoeuvring in complex environments, this could potentially further reduce their foraging performance. To what extent *Varroa* infestations affect flight control and associated learning in freely flying bees is, however, unknown.

In this study, we tested how the forager workforce from *Varroa*-infested colonies and control colonies adapt their flight behaviour to a challenging and unfamiliar environment. Hereby, we aimed to answer two main research questions: (i) *What is the effect of Varroa infestation of a colony on flight control of the colony's forager workforce?* (ii) *What is the functional biomechanical mechanism that causes the potential reduction in flight control of Varroa-infested colonies?* For this, we used stereoscopic videography analyses to study how forager bees from *Varroa*-infested colonies and control colonies land in challenging conditions, i.e. on a vertical platform in an unfamiliar environment. We performed this study at the colony level, without distinguishing between individual bees. Our results show that—in contrast with the forager bees of control colonies—the forager workforce of *Varroa*-infested colonies does not improve its landing performance over time, suggesting a reduction in skill-learning capabilities.

# 2. Material and methods

## 2.1. Experimental animals: *Varroa*-infested and control honeybee colonies with sister queens

For this study, we used three pairs of western honeybee (*A. mellifera*) colonies, kept in 8-frame wooden hives (Simplex, Inbuzz v.o.f., The Netherlands) at the Wageningen UR bee research facilities (see electronic supplementary material, table S1). The queens of each colony pair were sisters, such that the genetic differences between the pairs were minimal. One colony of each pair was treated against *V. destructor* with oxalic acid in the broodless period; the other colonies received no mite treatment. Due to queen problems half way through the experiment, one of the colonies treated against *Varroa* was replaced with a new sister colony (electronic supplementary material, table S1).

Allocation of colonies to the *Varroa*-infested group (V+) and control group (V−) was done based on actual mite counts [17]. We determined the level of *Varroa* infestation for each colony after the July runs and before the August runs of the experiment. A sample of, on average, 230 bees were taken from each colony and frozen at −20°C. The bees were unfrozen, the mites were washed off the bees with water and detergent, and then counted as the number of mites per 100 bees (electronic supplementary material, table S1).

Experiments were performed for 6 days in mid-July and 6 days in mid-August, and each day a single colony was tested. Sister colonies were tested during consecutive days, and the order of testing differed between July and August (electronic supplementary material, table S1).

## 2.2. Experimental set-up: flight arena with stereoscopic high-speed videography system

Experiments took place in a custom-built flight arena. The flight arena consisted of a $1 \times 1 \times 1$ m flight room, a horizontal feeding platform and a vertical landing platform to which a beehive could be connected. The beehive was positioned outside the arena. The feeding platform contained a honey solution (diluted $1:4$). A stereoscopic high-speed videography system was used to track the three-dimensional flight trajectories of bees landing on the vertical landing platform, when returning to the hive.

The videography system consisted of two synchronized high-speed cameras (Mikrotron EoSens CL MC1362, Mikrotron GmbH, Germany) with 50 mm Nikkor f1.8 AF-D (aperture set at f8) lenses, recording at 300 frames per second (exposure 1 ms, resolution $970 \times 1020$ pixels). The cameras were positioned on the side and bottom of the arena, and an area-of-interest of approximately $20 \times 20 \times 20$ cm in front of the landing platform was filmed. The camera images were backlit using two 45 W broad-spectrum LED light panels (3680 lumens, cold white 4500 K, 216 SMD2385 LEDs), positioned outside the flight arena opposing each camera. With this backlighting, the cameras viewed the flying bees as shadows in front of a bright background (figure 1*b,c*; electronic supplementary material, Movies S1 and S2).

We used the direct linear transformation (DLT) method [18] to convert corresponding points in the two images into estimates of three-dimensional positions [18]. For calibration we used 40 lead beads with known three-dimensional positions, randomly distributed throughout the region-of-interest. Positions of the corresponding lead beads in both views were manually indicated. DLT calibration was done each experimental day. As part of the DLT calibration routine, we estimated the reconstruction error for the digitized beats in all calibrations as $0.19 \pm 0.06$ mm (mean ± standard deviation, $n = 11$ calibrations).

The arena was placed inside a building such that climate and light conditions could be controlled. Temperature was controlled using a mobile air-conditioning system; during the experimental days, temperature and relative humidity were $27.7 \pm 3.4°C$ and $71 \pm 10\%$, respectively. The LED light panels that provided illumination for the cameras also provided visible light for the bees. To allow for proper in-flight orientation, four 15 W 368 nm UV blacklights (Sylvania F15 W T5 BL368, Osram USA) were placed on top of the flight arena, and the arena top panel was made of 3 mm thick UV-transparent polymethyl methacrylate (PMMA). To provide visual feedback for navigation and orientation, the side walls were covered with a random pixel array ($1 \times 1$ cm, random greyscale values from black to white). With a 2° angular sensitivity of the honeybee visual system [19], a honeybee should be able to detect a single $1 \times 1$ cm square at a distance of approximately 30 cm.

The vertical landing platform consisted of a white circular flat plate (12.5 cm diameter) with a hole in the middle. A tube connected the centre hole to the hive entrance, located outside the arena. To emphasize potential differences in landing performance, the platform was made of the white and slippery material Teflon. The homogeneous white platform provided low levels of visual contrast, and the slippery material challenged the motor system when landing.

## 2.3. Experimental procedure: recording landing manoeuvres of foraging honeybees

At the start of each experimental day, a hive was brought into the experimental room and connected to the flight arena. After an hour of acclimation (around 11.00), the bees were allowed to forage ad libitum by flying back and forth between the hive and food source. The system continuously stored video data into a buffer of 3 s. When the experimenter observed a landing attempt, the recording was stopped and the last 3 s of video was saved to a disk, after which the recording was restarted. Landing attempts were defined as flights in which the bees approached and touched the landing platform.

The experiment was stopped after approximately 5 h of measurements. After recording was stopped, we slowly reduced the intensity of the visible lights in the set-up, encouraging the bees to return back to the hive. After that, the hive was closed and moved back to its field location.

The foraging conditions in the experimental set-up were very different from those in the field location. Particularly the vertical white Teflon landing platform differed from the horizontal landing

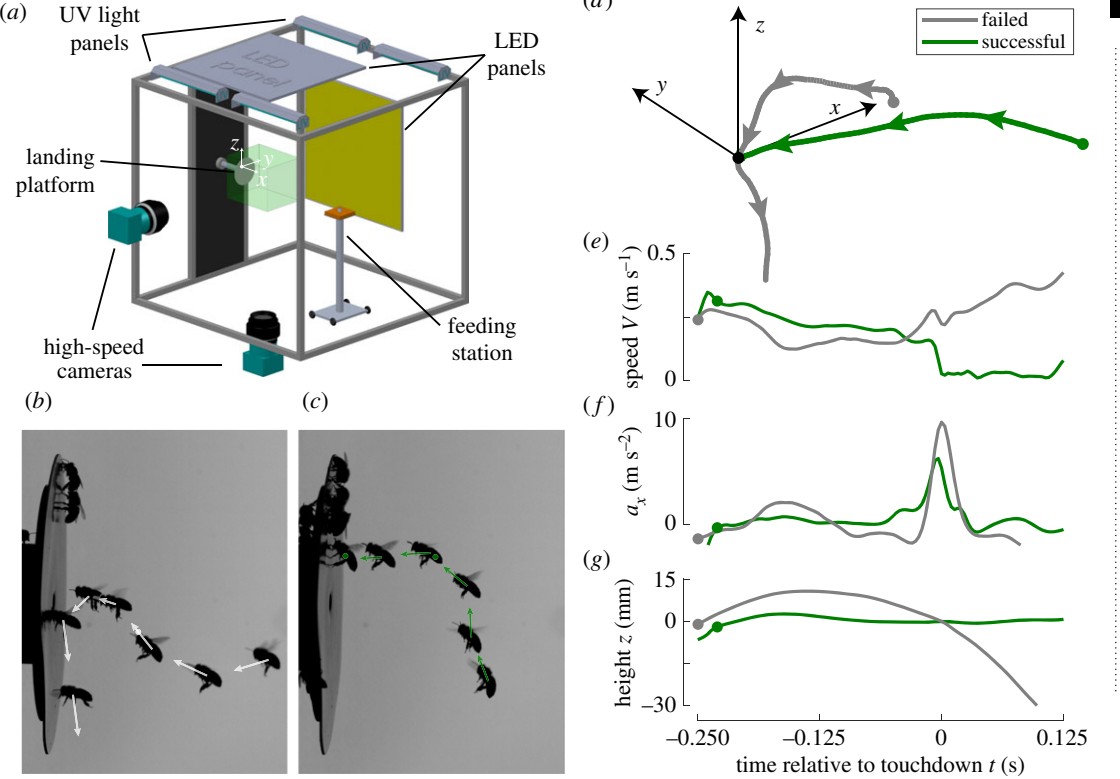

**Figure 1.** (a) The experimental set-up and (b–g) analysis approach applied to a failed landing without pre-touchdown leg extension (grey), and a successful landing with pre-touchdown leg extension (green). (a) The experimental set-up with a stereoscopic high-speed camera system, a feeding station and a vertical landing platform. The vertical landing platform was connected to a beehive placed outside the arena (not shown). The green box around the vertical landing platform depicts the viewing domain of the cameras. The Cartesian coordinate system has its origin at the landing point, the x-axis is perpendicular to the landing platform, and z is vertical up. (b,c) Photo montage from the side-view video of a failed landing (b) and a successful landing (c). The bee is shown at every 30th video frame ($\Delta t = 0.1$ s), and arrows depict in-plane velocity vectors. (d) Three-dimensional trajectories of both landings (all coordinate axes are 25 mm long). (e–g) The temporal dynamics of flight speed (e), acceleration normal to the landing platform, whereby positive values represent decelerations of the approaching bees (f) and height (g) throughout both manoeuvres. Grey and green dots show the point at which the bee enters the 50 mm sphere around the landing point (electronic supplementary material, *Methodology*). The landing point at $X = 0$ mm and time $t = 0$ s, is depicted by the black dot in (d).

plate of the wooden Simplex hives used in the field. Because the first and second trials for each colony were more than 41 days apart (electronic supplementary material, table S1), we assumed that the environment of the flight arena, and particularly that of the landing platform, was unfamiliar at the beginning of each experimental day.

## 2.4. Automatic and manual tracking of landing manoeuvres

We tracked the three-dimensional flight movements of honeybees using our in-house developed insect flight tracking algorithm. Based on machine-vision algorithms in OpenCV [20], we tracked the centroids of all bee silhouettes. From the collection of flight tracks in each camera view, we selected the ones approaching the landing platform simultaneously in both cameras, and reconstructed the three-dimensional track using DLT [18]. We then used a Kalman filter to remove tracking noise and to determine the velocity and acceleration vectors. The measurement noise covariance matrix of the Kalman filter was set to identity and the cross-product of the error covariance matrices to zero. The process noise matrix parameters were set to 0.5, 0.1 and 0.01 for position, velocity and acceleration, respectively.

The resulting temporal dynamics of position ($\mathbf{X}(t)$), velocity ($\mathbf{U}(t)$) and acceleration ($\mathbf{A}(t)$) were expressed in a Cartesian coordinate system with its origin at the position of first contact with the

landing platform, its $x$-axis oriented normal to the landing platform and $z$-axis vertical up (figure 1$a$; see electronic supplementary material for a list of symbols and abbreviations). The $z$-coordinate thus equals the height of the bee relative to its point of landing. For landing kinematic analyses (figure 1$d$–$g$), we used time relative to the moment of touchdown ($t = 0$ s at first contact); to analyse the change in behaviour throughout the day, we used time of day $t_{day}$, measured relative to the start of the experiment (11.00).

We manually determined the video frame of touchdown, whether a landing was successful and whether the animal extended its legs prior to touchdown. The touchdown was defined as the moment when the first body part touched the landing platform. In a successful landing the animal remained attached to the platform after first contact. Landing attempts in which bees flew away after touchdown, or initiated a second landing approach, were scored as failures. We scored leg extension at touchdown binomially as either absent ($L = 0$) or present ($L = 1$). See electronic supplementary material, Movies S1 and S2 for examples of a failed landing without leg extension and a successful landing with leg extension prior to touchdown, respectively.

## 2.5. Mechanistic model for landing success

To study how landing success is related to variations in flight kinematics throughout the landing manoeuvre, we developed a biomechanical mechanistic model for predicting landing success. To successfully land, a bee uses its sensory-motor control system to reduce its flight speed at free flight down to zero after touchdown [15]. The visual system provides most of the sensory input, and the wings and legs generate the motor output. Wing movements control in-flight braking and the legs can be used for braking at impact [21]. In-flight braking can be achieved by producing an aerodynamic force vector in the direction opposite to its flight speed (aerodynamic braking) and by converting kinetic energy into potential energy by gaining height. Braking at impact is achieved by absorbing the associated kinetic energy using the leg's muscle–tendon–cuticle system (i.e. the landing gear).

Therefore, we assume that the probability of making a successful landing decreases with approach speed ($V_{approach}$) and speed change at impact ($\Delta V_{impact}$), and that it increases with amount of leg extension prior to touchdown. Based on this, our mechanistic model for predicting landing success is

$$P(\text{success}) = f(V_{approach},\ \Delta V^*_{impact}, L), \tag{2.1}$$

where $P(\text{success})$ is the probability of making a successful landing and $\Delta V^*_{impact}$ is the speed change at impact relative to approach speed. Approach speed ($V_{approach}$) is defined as the average flight speed when within a 50 mm distance of the touchdown point. See electronic supplementary material, *Methodology* for details about the model and related metrics. Although relative leg-extension $L$ is a continuous parameter, we expressed it binomially as either absent ($L = 0$) or present ($L = 1$).

## 2.6. Analysis approach and statistics

All statistical analyses were performed using Matlab 2018b (Mathworks Inc.). Our analysis consisted of three steps:

Firstly, we tested whether landing success varied during the day and between the paired *Varroa*-infested and control sister colonies. For this, we used a generalized linear mixed-effects model (GLMM) with landing success as a binomial-dependent variable, and independent variables were time of day and *Varroa* infestation of the colony ($V-/V+$), including their interaction.

Secondly, we assessed how landing success depended on the landing kinematics, by fitting our mechanistic model (equation (2.1)) to the data of all analysed landing manoeuvres using a GLMM. In the GLMM, we set landing success as the binary dependent variable, and independent variables were approach flight speed, relative speed change at impact and pre-touchdown leg extension, including their two-way interactions.

Thirdly, we tested how the kinematics parameters that significantly affected landing success (based on the GLMM test of step two) varied as a function of time of day, and how this differed between the paired *Varroa*-infested and control sister colonies. For each test, the dependent variable was a kinematics parameter (approach flight speed, relative speed change at impact or pre-touchdown leg extension); independent parameters were always time of day and *Varroa* infestation ($V-/V+$), including their interaction.

For GLMM tests in which we found an effect of *Varroa* infestation on the dependent variable, we performed corresponding *post hoc* tests on both the *Varroa*-infested and control groups separately. For the effect of *Varroa* infestation on landing success, we also performed a *post hoc* test on each colony separately. For all *post hoc* tests, we used GLMMs with time of day as the independent variable.

For all GLMM tests, hive number and experimental day were set as random factors, and temperature at the moment of landing (including one-way interactions) was added as the covariate. An effect was assumed significant if the probability value (*p*-value) was smaller than 0.05. The minimal model was determined by iteratively removing the independent variables with the largest non-significant *p*-value; we always kept the independent variable of interest (*Varroa* infestation versus control) in the model. Statistical results are given as test statistics $t_{DF}$ (where DF is degrees-of-freedom) and *p*-value. For data exploration, we used pairwise linear correlation tests (electronic supplementary material, figure S1) and pairwise Wilcoxon rank sum tests (for non-normally distributed kinematics data). Unless explicitly stated otherwise, binary data were provided as *mean ± standard error* and non-normally distributed data as *median (first quartile–third quartile)*.

# 3. Results

## 3.1. *Varroa* infestation in the different colonies

Two out of the seven colonies showed mite infestations. In these *Varroa*-infested colonies, the average infestation rate was moderate [22], as it increased from 1.1 mites per 100 bees in early July to 2.0 in late August (electronic supplementary material, table S1). In the other colonies no mites were detected, in both July and August; they were therefore considered as the control colonies. The complete experimental dataset (electronic supplementary material, table S1), therefore, consisted of 4 days of experiments with two *Varroa*-infested colonies and 8 days of experiments with five control colonies (including the replacement hive). The dataset of paired *Varroa*-infested and control sister colonies consisted of the 8 experimental days of two *Varroa*-infested colonies and two paired control colonies (electronic supplementary material, table S1).

## 3.2. Landing success of bees from paired *Varroa*-infested and control sister colonies

Data for paired sister colonies included 233 analysed landing manoeuvres for the two *Varroa*-infested colonies and 174 landing manoeuvres for the paired control colonies (electronic supplementary material, Database S1). Of these 407 landings, 211 were scored as successful and 196 as failed, resulting in an average landing success percentage of 52%.

Based on these landing manoeuvres, we tested how landing success of forager bees changed with time of day, for *Varroa*-infested colonies and control colonies (figure 2; electronic supplementary material, table S2). The binomial GLMM results show that landing success increased with time of day ($t_{403} = 2.848$, $p = 0.005$) and that there is a significant interaction between time of day and *Varroa* infestation ($t_{403} = -2.282$, $p = 0.023$). This suggests that time of day has a different effect on *Varroa*-infested colonies and control colonies. The *post hoc* test on the control colonies showed an increase in landing success with 10% h$^{-1}$ ($t_{172} = 2.848$, $p = 0.005$), reaching 80% at the end of a 5 h day. By contrast, the *post hoc* test on *Varroa*-infested colonies showed that for these bees landing success did not change significantly with time of day ($t_{231} = 0.268$, $p = 0.789$), and thus they remained at an approximately constant success rate of 44%. The *post hoc* test on the paired sister colonies separately (electronic supplementary material, figure S2 and table S2) showed similar trends in landing success with time of day as for all data combined.

## 3.3. The biomechanics of successful and failed landing manoeuvres

To determine what metrics affected landing success, we comparatively analysed the biomechanics of all 263 successful and 292 failed landing manoeuvres (electronic supplementary material, Database S1, table S1 and figure 3).

On average, the bees approached the landing platform at roughly a constant flight speed (figure 3*b*), and at touchdown ($t = 0$ s) the bees quickly decelerated to reduce the flight speed to values close to zero

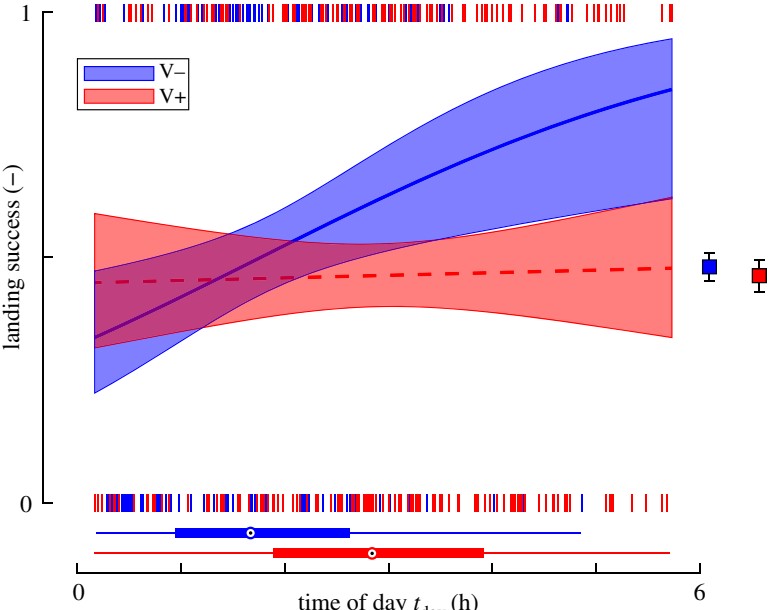

**Figure 2.** The temporal dynamics of landing success probability throughout an experimental day shows that the forager workforce of the paired control colonies (V−) increase landing success during the day, whereas bees from the paired *Varroa*-infested colonies (V+) did not. Red data are of bees from paired *Varroa*-infested sister colonies ($N_{V+}$ = 233 landings) and blue data are of paired control colonies ($N_{V−}$ = 174 landings). The short vertical lines at zero and one show failed and successful landings, respectively. The trend lines show predicted success probability and 95% confidence interval (electronic supplementary material, table S2). Box plot at the bottom shows the distribution of landings throughout the day. Symbols on the right shows mean ± standard error of all analysed landings.

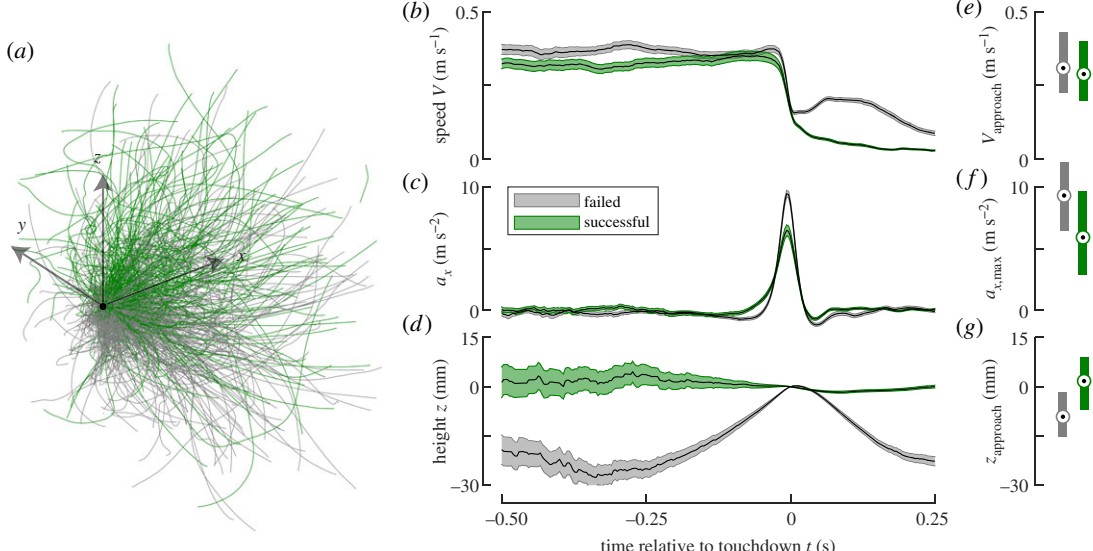

**Figure 3.** The flight kinematics of all analysed landing manoeuvres. Data in grey and green are of failed and successful landings, respectively. (*a*) Three-dimensional trajectories of all landings (system axes are 50 mm). (*b*–*d*) Temporal dynamics of flight speed (*b*), acceleration normal to the landing platform, whereby positive values represent decelerations of an approaching bee (*c*) and flight height (*d*). The trend lines depict the mean ± standard error of that group at each recorded video frame. (*e*–*g*) Median (second quartile–third quartile) of approach flight speed (*e*), maximum impact acceleration normal to the landing platform (*f*) and approach flight height (*g*) of successful and failed landings. (*a*–*d*) $N_{successful}$ = 263 landings; $N_{fail}$ = 293 landings. (*e*–*g*) $N_{successful}$ = 217 landings; $N_{fail}$ = 269 landings.

(figure 3*b,c*). For failed landings, the speed went up again after touchdown (electronic supplementary material, Movie S1); for a successful landing, the speed of the bee reduced slowly to zero, when the animal settled on the platform (electronic supplementary material, Movie S2).

Although the temporal dynamics of flight speed seemed to differ between failed and successful landings (figure 3b), the average approach speeds ($V_{approach}$) were not significantly different (figure 3e; $V_{approach,successful} = 0.29$ (0.20–0.40) m s$^{-1}$, $N_{successful} = 217$ landings; $V_{approach,failed} = 0.31$ (0.22–0.43) m s$^{-1}$, $N_{failed} = 269$ landings; $p = 0.068$).

The accelerations normal to the platform ($a_x$) were close to zero throughout the approach phase of the landing, but they rapidly increased at touchdown ($t = 0$ s, figure 3c). This increase corresponds to reductions of the approach speeds relative to the platform (see the coordinate system in figure 1a). The median peak acceleration ($a_{x,max}$) is 58% higher for the failed landings, compared to successful landings (figure 3f; $a_{x,max,successful} = 5.9$ (2.9–9.7) m s$^{-2}$; $a_{x,max,failed} = 9.3$ (6.4–12.0) m s$^{-2}$; $p < 0.001$).

For failed landings, bees approached the landing platform on average from below, and rapidly increase their height just prior to touchdown (figure 3d,g). For successful landing, this increase in height was absent. As a result, the average approach height ($z_{approach}$) was significantly lower for the failed landings compared to the successful ones (figure 3g; $z_{approach,successful} = 1.7$ (−6.9 to 9.1) mm; $z_{approach,failed} = −9.2$ (−15.3 to −1.7) mm; $p < 0.001$).

Based on the accelerations at impact and the approach speeds, we estimated the relative speed changes at impact ($\Delta V^*_{impact}$, electronic supplementary material, equation (S3)). These were significantly higher for failed landings compared to successful ones ($\Delta V^*_{impact,successful} = 0.21$ (0.13–0.28), $N_{successful} = 217$ landings; $\Delta V^*_{impact,successful} = 0.28$ (0.22–0.36), $N_{failed} = 269$ landings; $p < 0.001$). This shows that for successful landings, bees had to reduce their flight speed at impact with 21% of the total speed change, whereas for failed landing this was 28%.

For 201 of all landing manoeuvres, bees extended their legs prior to touchdown and for 285 landings they did not. Manoeuvres with leg extension led in 84% to a successful landing, whereas only 17% of the manoeuvres without pre-touchdown leg extension were successful (electronic supplementary material, Database S1).

## 3.4. The mechanistic model for predicting landing success based on landing kinematics

These findings suggest that the parameters defined in equation (2.1) do affect landing success probability, which we subsequently tested using the GLMM (figure 4; electronic supplementary material, table S3). The results showed significant interactions between the approach speed and leg extension and between relative speed change at impact and leg extension (electronic supplementary material, table S3); the interaction between approach speed and relative impact speed change was not significant and was therefore removed from the model.

The GLMM results (figure 4; electronic supplementary material, table S3) confirm that bees that extend their legs prior to touchdown have a much higher chance of making a successful landing (figure 4b) than bees that do not (figure 4a). Within the parametric space as depicted in figure 4a, the model predicts that without leg extension bees have on average an 12% chance of making a successful landing (figure 4a). This probability decreases with an increase in both approach speed (figure 4c, $L = 0$) and relative speed change at impact (figure 4d, $L = 0$).

The probability of making a successful landing is much higher for landings with leg extension prior to touchdown (figure 4b, approx. 63% within the parametric space). With leg extension, the predicted chance of making a successful landing depends on approach speed in an unexpected way (figure 4c, $L = 1$): bees show a weak tendency to make more successful landings if they approach the platform with a higher flight speed. Relative speed change at impact shows a much stronger effect (figure 4d, $L = 1$). At an impact speed change of zero, the model predicts that 98% of the landings are successful. This percentage rapidly drops to 19% at the highest recorded value of relative speed change ($V^*_{impact} = 0.66$).

## 3.5. Variations of landing kinematics throughout the experimental day

The GLMM test showed that all three tested kinematics metrics significantly affect landing success (figure 4; electronic supplementary material, table S3). Therefore, we used GLMMs to test how each kinematics parameter varied with time of day, and how this differed between bees from the paired *Varroa*-infested and control sister colonies (figure 5; electronic supplementary material, table S3).

Pre-touchdown leg extension varies significantly with time of day (figure 5a and electronic supplementary material, table S4, $t_{404} = 2.577$, $p = 0.010$), and between bees from *Varroa*-infested colonies and control colonies ($t_{404} = −2.093$, $p = 0.037$). The interaction between time-of-day and *Varroa* infestation was not significant and therefore dropped from the minimal model. The resulting model predicts that for all paired sister colonies, during a day of foraging the percentage of leg extension

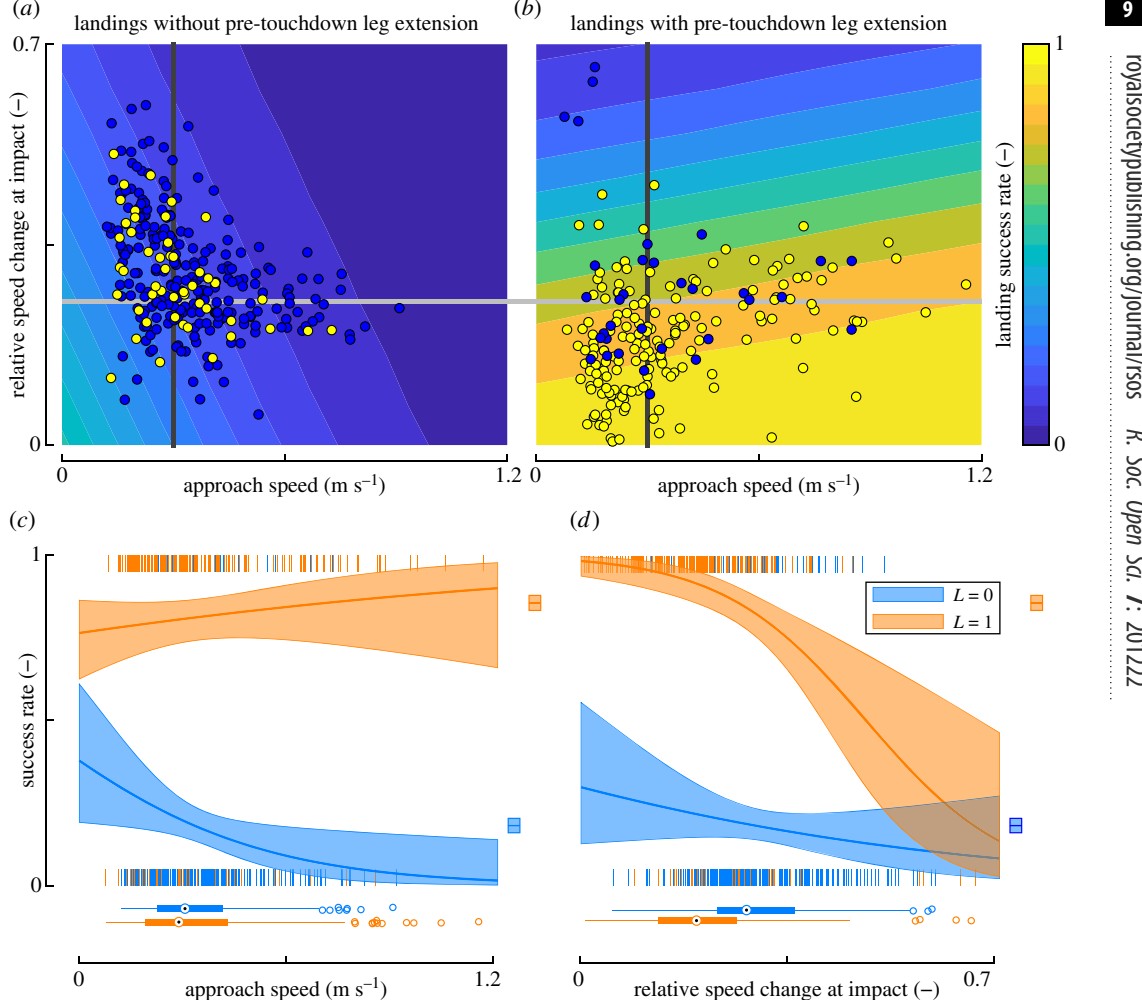

**Figure 4.** Landing success depends on leg extension prior to touchdown, approach flight speed and relative speed change at impact. (*a,b*) The success rate predicted using our mechanistic landing model (equation (2.1) and electronic supplementary material, table S3) throughout the parametric space of $V_{approach}$ and $V^*_{impact}$, for landings without (*a*) and with (*b*) pre-touchdown leg extension. Data points are of all successful landings (yellow) and failed landings (blue) within each leg-extension group (*a*: $N_{successful} = 48$ landings and $N_{failed} = 236$; *b*: $N_{successful} = 169$ and $N_{failed} = 33$). (*c*) Success probability predicted by the GLMM at variable $V_{approach}$ and constant $V^*_{impact} = 0.25$, in the sub-space highlighted by the light grey line in (*a,b*). (*d*) Predicted success probability at variable $V^*_{impact}$ and constant $V_{approach} = 0.3$ m s$^{-1}$, in the sub-space highlighted by the dark grey line in (*a,b*). (*c,d*) Predicted mean and 95% confidence interval of success probability for landings without (blue) and with (orange) pre-touchdown leg extension. Small vertical blue ($L = 0$) and orange lines ($L = 1$) at zero and one depict recorded landings; squares on the right depict means ± standard errors for these landings; box plots on the bottom show their distributions.

increased by 5% h$^{-1}$ (figure 5*a*). Hereby, the leg extension percentage of the control colonies increased from 38% at the start of the day to 63% after 5 h. The *Varroa*-infested colonies had equivalent percentages of 28% and 52% at the start and end, respectively (figure 5*a*).

The approach speed did not vary with time of day (figure 5*b*, electronic supplementary material, table S4, $t_{350} = -0.285$, $p = 0.776$) nor between bees from *Varroa*-infested colonies and control colonies ($t_{350} = 0.212$, $p = 0.832$).

The relative speed change at impact did vary significantly with time of day (figure 5*c*; electronic supplementary material, table S4, $t_{349} = -4.697$, $p < 0.001$) and between *Varroa*-infested colonies and control colonies ($t_{349} = -3.022$, $p = 0.003$). Here, the interaction between time of day and *Varroa* infestation was also significant ($t_{350} = 4.118$, $p < 0.001$), suggesting that variations of relative speed change at impact during the day differed between bees from *Varroa*-infested colonies and control colonies. A *post hoc* test on data of control colonies shows that control bees significantly reduced

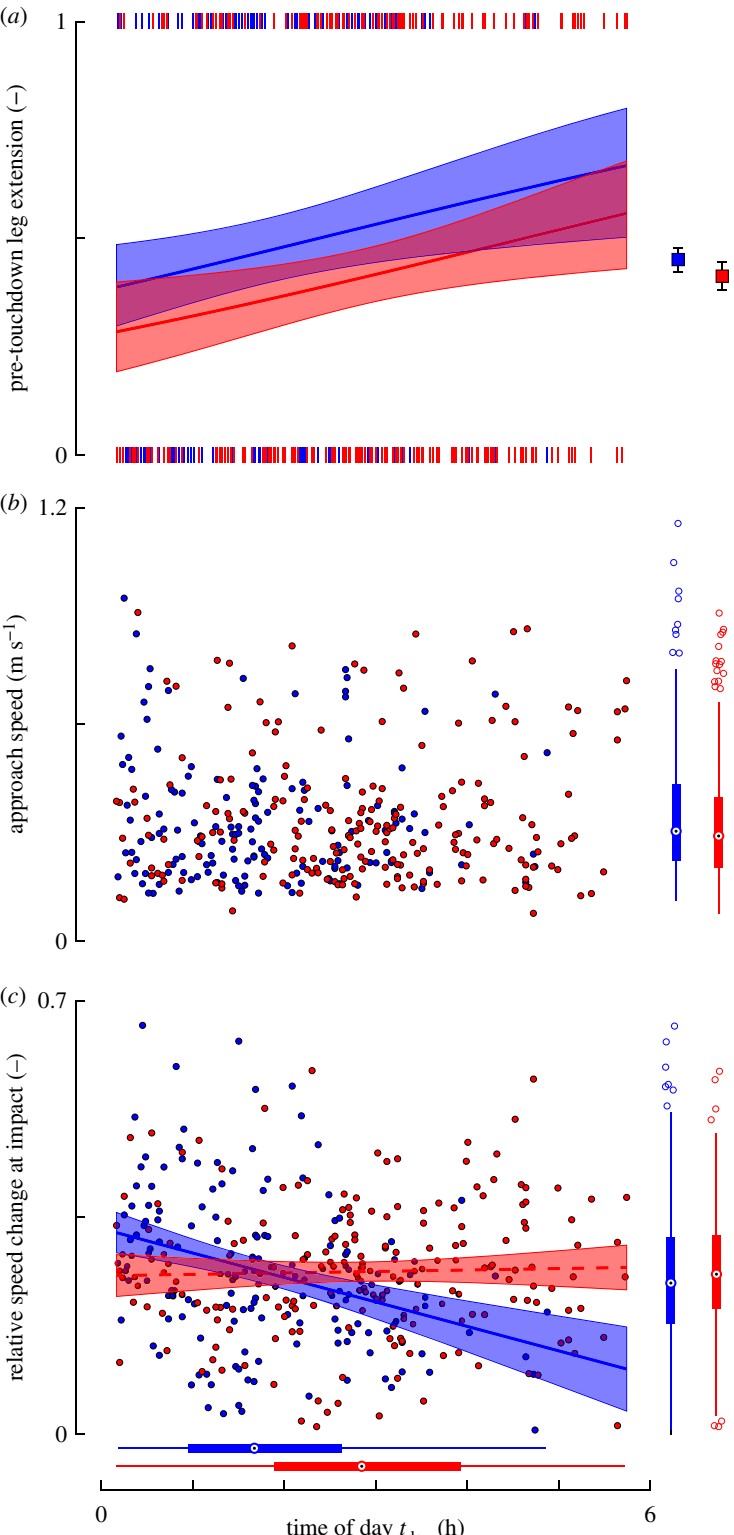

**Figure 5.** Variation throughout the day of the kinematics parameters that affect landing success. (*a*) During a day of foraging, bees increase their rate of pre-touchdown leg extension; (*b*) throughout the day, bees did not change the approach flight speed; (*c*) during a foraging day, bees from control colonies decreased the relative speed change at impact, whereas bees from *Varroa*-infested colonies did not do this. All data shown are of bees from paired control sister colonies (blue, $N_{V-} = 174$ landings) and paired *Varroa*-infested sister colonies (red, $N_{V}+ = 206$ landings). (*a*,*c*) The shaded trend lines show temporal dynamics of mean and 95% confidence intervals predicted by the GLMM (electronic supplementary material, table S4); (*a*) Small vertical lines at zero and one indicate landings at which the animal did not ($L = 0$) or did ($L = 1$) extend its legs prior to touchdown, respectively; the squares on the right of (*a*) depict means ± standard errors; (*b*,*c*) each data point shows the result of a single landing; the box plots below and on the right of (*b*,*c*) show distributions of all data.

speed change at impact from 33% of the average flight speed at the start of an experiment to 13% after 5 h (electronic supplementary material, table S4, $t_{145} = -4.356$, $p < 0.001$). This corresponds to an hourly reduction in the impact of 4%. By contrast, the *post hoc* model on data of *Varroa*-infested colonies showed no significant changes in relative speed change at impact with time of day (electronic supplementary material, table S4, $t_{204} = 0.436$, $p = 0.663$).

# 4. Discussion

## 4.1. Forager bees of *Varroa*-infested colonies do not improve landing success during a day of foraging

Here, we studied how the forager workforce from *Varroa*-infested colonies and control colonies performed landing manoeuvres in an environment that was unfamiliar at the beginning of each experimental day. Control colonies showed increased landing success in the course of the day (from 32% to 80% successful landings) due to changed landing kinematics. By contrast, forager bees from *Varroa*-infested colonies did not show this increase in performance but remained at an average success percentage of 46% throughout a day of foraging. Our results suggest that control bees were able to successfully adapt their landings to the new unfamiliar situation [23]. A lack of this adaptive capability in bees from *Varroa*-infested colonies suggests that the previously observed reduced learning ability in *Varroa*-infested bees [6] also affects the sensory-motor flight control system of freely flying forager bees.

## 4.2. Bees increase landing success by improving their flight control

To understand how forager bees use their sensory-motor flight control system to perform successful landings, we analysed the kinematics of failed and successful landings of these bees using our mechanistic model (equation (2.1)). The model estimates the probability of a successful landing based on approach speed, relative speed change at impact and leg extension prior to touchdown. Results showed that landing success depended strongly on pre-touchdown leg extension, with 84% success with leg extension and only 17% without leg extension. This confirms the importance of leg movement control for successful landings in honeybees reported previously [15].

Landing success depended only weakly on the approach flight speed (figure 4). For bees that did not extend their legs prior to touchdown, landing success rate increased with decreasing approach speed, which was the expected relation. Surprisingly, for bees extending their legs prior to touchdown, an opposite relationship was observed. This counterintuitive result suggests that more experienced bees that were able to perform more controlled landings, also flew faster, allowing them to forage more quickly and more efficiently. This is in line with recent finding that more experienced bees collect more food per trip and also exhibit a higher foraging frequency [23].

The relative change in flight speed upon impact also corresponded significantly to landing success. This metric ($\Delta V^*_{impact}$, electronic supplementary material, equation S3) quantifies how much a bee brakes at impact, relative to its in-flight braking prior to impact. As expected, landings with a relatively high change in speed at impact had a lower success rate. This effect was most prominent for bees that extended their legs prior to touchdown. In fact, for these landings, the success rate was almost 100% at the lowest recorded speed changes at impact, but the success rate reduced to 19% at the highest recorded impacts. To minimize the relative impact at touchdown, the animal needs to maximize in-flight braking using height gain or aerodynamic braking. Surprisingly, bees that performed successful landings did not gain height, whereas failed landings did show a height increase prior to touchdown (figure 3*c,f*). We cannot explain this unexpected result, but it does suggest that bees that performed successful landings used active aerodynamic braking instead of gaining height.

These combined results thus show that landing success is the highest when a landing bee both maximizes its in-flight aerodynamic braking and extends its legs prior to touchdown. By contrast, the approach flight speed prior to landing has a much smaller and sometimes counterintuitive positive effect on landing success. This shows that landing success primarily depends on *flight control* parameters and much less on flight speed as a metric for *flight performance*.

## 4.3. Forager bees from *Varroa*-infested colonies fail to improve in-flight aerodynamic braking

During a day of foraging in an unfamiliar environment, the forager workforce of *Varroa*-infested colonies change their landing behaviour differently from control bees. Bees from *Varroa*-infested colonies foraging in an unfamiliar environment only increased their pre-touchdown leg extension rate during the day, but that alone did not significantly increase their landing success. By contrast, control bees that foraged in the same unfamiliar environment, improved their manoeuvre control throughout the day by both increasing the pre-touchdown leg extension rate and performing more in-flight braking. They did not reduce their approach flight speed, which allowed them to improve their landing success rate without having to increase the duration of the landing manoeuvre.

The sensory-motor system that controls these landing manoeuvres consists of a sensory input system (primarily visual), a neural processor and two motor output systems. The wing motor system controls the approach flight dynamics, and the leg motor system controls leg extension, braking at touchdown and attachment. In free flight experiments, it is difficult to determine which of these sub-systems is most affected by *Varroa* infestation. Still, our finding that bees from infested colonies did improve on the leg extension rate during the day, but did not improve their in-flight braking before impact, suggests that primarily the flight motor system is affected.

## 4.4. Causes and consequences of the *Varroa*-induced reduction in flight control

The literature suggests several different causes for the absence of flight control adaptability in *Varroa*-infested colonies: (i) It might be due to impaired cognitive function of individual bees that were infected by *V. destructor* during pupal development [6]. (ii) It might be due to the *Varroa*-induced change in forager workforce demographics [7,23]. The younger forager bees show reduced forager efficiency [23], and thus might also have impaired flight skill-learning capabilities. (iii) A third indirect cause may have been diseases transmitted by *V. destructor* mites such as the deformed wing virus. Active foragers without visual signs of the disease can have reduced flight duration and flight distance [12], and thus also their flight control might have been compromised.

Our finding that *Varroa* infestation affected wing motor control and not leg motor control supports a role for covert deformed wing virus infections [12]. Because the *Varroa*-infestation rates in our infested colonies were moderate (1–2%), only a few of the individual bees were most likely infected during development. This suggests that impaired cognitive function as a result of *Varroa* infestation during pupation is most likely not the cause of the colony-wide absence of improvement in landing control. Additional research is needed to test whether *Varroa*-induced changes in forager workforce demographics contribute to the reduction in flight control of forager bees.

Even though *Varroa*-infestation rates were moderate (1–2%), the likelihood of these infested colonies to survive winter would have already been compromised [22]. Based on these infestation rates and an assumed doubling of mites between August and September, the likelihood of colony collapse during the following winter would have been 25–60%.

The flexibility in flight control behaviour, as shown for control bees, might be important for foraging in a large variety of natural conditions. Such flexibility in flight control is expected to be particularly relevant when weather conditions deteriorate or when bees need to switch foraging between different flower patches or flower species throughout the season. By contrast, bees from *Varroa*-infested colonies were unable to improve on their landing skills during at least the first 5 h in an unfamiliar environment, potentially compromising their capability to adapt to varying foraging conditions. Therefore, the here-identified compromised landing skills of the forager workforce of *Varroa*-infested colonies may help explain their impaired homing ability [24] and reduced forager efficiency [23].

Data accessibility. The data that supports this study, including all analysed flight trajectories, derived metrics and meta-data, are available in electronic supplementary material, Database S1.

Authors' contributions. F.T.M., C.v.D. and F.V.L. conceived the study. F.T.M., C.v.D., H.L. and F.V.L. designed the experiment. H.L. performed the experiments with support of C.v.D.; M.L. developed the video tracking software. H.L. performed the video tracking with support from M.L.; F.T.M. and H.L. analysed the data with support of L.J.d.V.; F.T.M. wrote the original draft of the manuscript. All authors contributed in writing the manuscript and gave final approval for publication and agree to be held accountable for the work performed therein.

Competing interests. We have no competing interests.

Funding. L.J.d.V. is supported by a fellowship from the Wageningen Institute of Animal Sciences (WIAS).

Acknowledgements. We thank Remco Pieters for his tech-support and Vincent van de Pas for assisting in video analyses. We thank Inbuzz v.o.f. for taking care of the colonies, and all technicians of Bees@wur for supporting the fieldwork.

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
