## [Reviewer comments · Royal Society Open Science]

Review History

RSOS-201222.R0 (Original submission)

Review form: Reviewer 1

Is the manuscript scientifically sound in its present form?

Yes

Are the interpretations and conclusions justified by the results?

Yes

Is the language acceptable?

Yes

Do you have any ethical concerns with this paper?

No

Have you any concerns about statistical analyses in this paper?

No

Recommendation?

Accept as is

Comments to the Author(s)

I appreciate the careful responses the authors made to my comments and concerns. The authors have sufficiently addressed these and I am happy to recommend the paper to be accepted for publication.

Review form: Reviewer 2

Is the manuscript scientifically sound in its present form?

Yes

Are the interpretations and conclusions justified by the results?

Yes

Is the language acceptable?

Yes

Do you have any ethical concerns with this paper?

No

Have you any concerns about statistical analyses in this paper?

No

Recommendation?

Accept as is

Comments to the Author(s)

I am satisfied with the authors' edits addressing my concerns about the paper. Well done.

Decision letter (RSOS-201222.R0)

Dear Dr Muijres,

It is a pleasure to accept your manuscript entitled "*Varroa destructor* infestation impairs the improvement of landing performance in foraging honeybees" in its current form for publication in Royal Society Open Science. The comments of the reviewers who reviewed your manuscript are included at the foot of this letter.

Additionally, we note that the following email address is marked as 'invalid'. Please kindly provide the correct email address for the following co-author, in reply to this email:

heleen.lugt@wur.nl

Best regards,

on behalf of Dr Jake Socha (Associate Editor) and Professor Kevin Padian (Subject Editor).

Associate Editor Dr Jake Socha Comments to Author:

The previous reviewers (from the previous submission at Proceedings B) were both fully satisfied with the new revisions in response to their comments. This paper is a solid contribution that improves our understanding of the biomechanics and ecological context of flight in bees. Congratulations!

Reviewer(s)' Comments to Author:

Reviewer: 1
Comments to the Author(s)

I appreciate the careful responses the authors made to my comments and concerns. The authors have sufficiently addressed these and I am happy to recommend the paper to be accepted for publication.

Reviewer: 2
Comments to the Author(s)

I am satisfied with the authors' edits addressing my concerns about the paper. Well done.
